# Disseminated Human Subarachnoid Coenurosis

**DOI:** 10.3390/tropicalmed7120405

**Published:** 2022-11-29

**Authors:** Jason Labuschagne, John Frean, Kaajal Parbhoo, Denis Mutyaba, Tanyia Pillay, Shareen Boughan, Hlezikuhle Nkala

**Affiliations:** 1Department of Neurosurgery, University of the Witwatersrand, Johannesburg 2193, South Africa; 2Department of Paediatric Neurosurgery, Nelson Mandela Children’s Hospital, Parktown, Johannesburg 2193, South Africa; 3Centre for Emerging Zoonotic and Parasitic Diseases, National Institute for Communicable Diseases, Division of the National Health Laboratory Service, Johannesburg 2193, South Africa; 4Wits Research Institute for Malaria, University of the Witwatersrand, Johannesburg 2193, South Africa; 5Department of Paediatrics and Child Health, University of the Witwatersrand, Johannesburg 2193, South Africa; 6Department of Paediatric Neurology, Nelson Mandela Children’s Hospital, Johannesburg 2193, South Africa; 7Department of Radiology, Nelson Mandela Children’s Hospital, Johannesburg 2193, South Africa; 8Department of Radiology, University of the Witwatersrand, Johannesburg 2193, South Africa

**Keywords:** coenurosis, neurocysticercosis, *Taenia solium*, *Taenia serialis*

## Abstract

Background: Traditionally, human coenurosis has been ascribed to *Taenia multiceps* while neurocysticercosis has been attributed solely to *Taenia solium* infection. Historically, however, the identification and differentiation of cestodal infection was primarily based on inaccurate morphological criteria. With the increasing availability of molecular methods, the accuracy of identification of the larval cestode species has improved, and cestodal species not typically associated with central nervous system (CNS) infection are now being identified as aetiological agents. Case report: We present a case of a 5-year-old male patient who presented with acute hydrocephalus. Initial MRI revealed multiple cysts in the cerebrospinal fluid (CSF) spaces with a predominance of clumped grape-like cysts in the basal cisterns with resultant acute obstructive hydrocephalus. The child underwent an emergency ventriculo-peritoneal (VP) shunt. A presumptive diagnosis of neurocysticercosis racemosus was made and the child was started on empiric albendazole (15 mg/kg/day) and praziquantel (30 mg/kg/day) treatment, along with concomitant prednisone (1 mg/kg) treatment. Despite prolonged anti-helminthic therapy, the child continued to deteriorate, and endoscopic removal of the 4th ventricular cysts was required. Post-operative MRI revealed radiological improvements, with a reduction in the number and size of cysts, especially in the basal cisterns, with no cysts visualized in the fourth ventricle. DNA was extracted from CSF and cyst tissue using the QiAMP DNA mini kit (Qiagen). The PCR performed on the extracted DNA displayed a band of 275 bp on an agarose gel. The consensus sequence had 97.68% similarity to *Taenia serialis* 12S ribosomal RNA gene. The child, unfortunately, continued to do poorly, requiring multiple VP shunt revisions for repeated blockage of the VP shunt system, and ultimately demised, despite the ‘successful’ surgical intervention and continued maximal medical management. Discussion and conclusions: There have been approximately 40 reported cases of human CNS coenurosis, with the assumed etiological agent being confined to *T. multiceps*. In 2020, the first case of human CNS coenurosis caused by *T. serialis* was reported. This case involved a single parenchymal lesion in the occipital lobe, which, following complete surgical excision, was confirmed to be *T. serialis* by mitochondrial gene sequencing. The case we present is the first case of disseminated subarachnoid coenurosis caused by *T. serialis*. It appears that *T. serialis* infection can mimic either of the two basic pathological forms of neurocysticercosis, namely, cysticercosis cellulosae or cysticercosis racemosus. We postulate that the term coenurosis racemosus is applicable if CNS *T. serialis* infection presents with extensive, multiple grape-like bladders proliferating within the subarachnoid space.

## 1. Introduction

Traditionally, human coenurosis has been ascribed to *Taenia multiceps,* while neurocysticercosis (NCC), in both cysticercosis cellulosae and cysticercosis racemosus forms, has been attributed solely to *Taenia solium* infection. Historically, the identification and differentiation of cestodal infection was primarily based on a potentially inaccurate morphological criteria such as larval hook size. With the increasing availability of molecular methods, the accuracy of identification of the larval cestode species has improved, and cestodal species not typically associated with central nervous system (CNS) infection are now being identified as aetiological agents. This report highlights only the second case of *T. serialis* CNS coenurosis and the first description of disseminated subarachnoid (racemose-like) coenurosis caused by this species.

## 2. Case Report

A 5-year-old male patient presented with a one-month history of intermittent vomiting, headaches, regression of milestones and recent onset seizures. The child had been living in an area with endemic *T. solium* taeniosis/cysticercosis in humans and pigs [1]. 

Initial MRI revealed multiple cysts in the cerebrospinal fluid (CSF) spaces with a predominance of clumped grape-like cysts in the basal cisterns with resultant acute obstructive hydrocephalus. The cysts demonstrated typical MRI signal intensity—T1 hypointense, T2 hyperintense with suppression on FLAIR. Some cysts demonstrated a central dot in keeping with a cestode scolex, typical of NCC (Figure 1). There were multiple cysts in the posterior spinal epidural space extending from the lower cervical to the lumbar region with smaller cysts within the subarachnoid space around filum terminale (Figure 2).

The child underwent an emergency ventriculo-peritoneal (VP) shunt. CSF was collected for biochemistry along with microscopy, culture and sensitivity, all of which was noncontributory. The child was started on empiric albendazole (15 mg/kg/day) and praziquantel (30 mg/kg/day) treatment, along with concomitant prednisone (1 mg/kg), for a week, followed by a tapering of the prednisone. 

The child had a relatively rapid response to treatment, his level of consciousness improved, and he was discharged home within seven days to complete a month’s course of both the albendazole and praziquantel. 

Following discharge, the child did well, with continued improvements in his symptoms at his clinic follow up visits. At three months post-discharge, however, the child presented with a depressed level of consciousness. Computerized tomography (CT) brain scan at this stage revealed acute hydrocephalus and the child underwent an emergency VP shunt revision. Once again, he responded well to treatment and was discharged within the week.

Again, after discharge, he was doing well at his follow-up clinic visits until, at approximately three months post second discharge, he required admission for headaches and persistent vomiting. He underwent two shunt revisions but, on both occasions, his shunt was found to be functional. He was started on repeat albendazole (15 mg/kg/day) and praziquantel at a higher dose (50 mg/kg/day) for an additional month of therapy, which he received in hospital. He also received high-dose intravenous dexamethasone (4 mg TDS), which was weaned to prednisone 1 mg/kg over a period of a month.

Follow-up MRI at six months demonstrated a redistribution of cysts with a slight decrease in size and number of cysts around the basal cisterns, an increase in number in the occipital horn of the left lateral ventricle and the fourth ventricle, and new cysts in the occipital and temporal horn of the right lateral ventricle. There was a new isolated right frontal lobe lesion that was hypointense on T2 with blooming on GRE, most likely indicating a calcified granuloma. (Figure 3)

Despite the high-dose antihelminth therapy, the child continued to deteriorate, developing intractable vomiting and subsequent cachexia. It was speculated that his persistent vomiting and subsequent continued weight loss was secondary to direct pressure from the cysts on the fourth ventricular floor. Given the child’s cachectic state, minimally invasive trans-ventricular, trans-aqueductal endoscopic aspiration of the cysts was favored over open surgical resection. A flexible endoscope (Karl Storz, Tuttlingen, Germany) was inserted into the right lateral ventricle and navigated through the dilated sylvian aqueduct. Once in the fourth ventricle, a combination of manual cyst retrieval with biopsy forceps and simple aspiration resulted in the removal of numerous cysts. By the end of the procedure, no cysts were visible within the fourth ventricle. After successfully clearing the fourth ventricle, only incomplete retrieval of cysts from the lateral ventricles was possible, as unlike in the fourth ventricle, where all the cysts were free-floating and non-adherent, a significant number of cysts within the lateral ventricles were tightly adherent to the ependymal lining and could not be safely removed. 

Following the transaqueductal cyst removal, the child had an almost immediate improvement in his vomiting and headaches. 

A follow-up MRI three weeks later showed improvements, with a reduction in the number and size of cysts, especially in the basal cisterns, with cysts no longer visualized in the fourth ventricle at all. (Figure 4).

The child, unfortunately, continued to do poorly, requiring multiple VP shunt revisions for repeated blockage of the VP shunt system, and ultimately demised, despite the ‘successful’ surgical intervention and continued maximal medical management.

## 3. Histopathological Diagnosis

The excised material was sent for histopathological diagnosis. The processed sections revealed the larval forms of an organism comprising scoleces and refractile hooklets. In addition, the larva form demonstrated duct-like invaginations and was lined by a double layer of eosinphilic membranes. As the clinical and radiological picture suggested racemose cysticercosis, a detailed morphological analysis of the surgically removed parasitic tissue arrangement was not requested, and a histopathological diagnosis of cysticercosis was reported. 

## 4. Genetic Analysis

DNA was extracted from CSF and cyst tissue using the QiAMP DNA mini kit (Qiagen, Hilden, Germany)) according to the manufacturer’s instructions. The cerebrospinal fluid was processed prior to DNA extraction by centrifugation at 800× *g* for 3 min. A cestode-specific polymerase chain reaction (PCR) was performed on extracted DNA using forward and reverse primers targeting a region of the 12S ribosomal RNA gene, as previously described [2]. The PCR reaction contained 1 × KAPA2G Robust HotStart ReadyMix (Roche Diagnostics, Midrand, South Africa), 0.5 µM of each primer and 5 µL of eluted DNA. Cycling conditions were 95 °C for 3 min, followed by 35 cycles of 95 °C for 3 min, 55 °C for 15 s and 72 °C for 15 s, followed by a final elongation step of 72 °C for 15 s. The PCR product was analyzed using agarose gel electrophoresis and GeneSnap software (Syngene, Bangalore, India). The unpurified PCR products were subjected to Sanger sequencing by Inqaba Biotec (Pretoria, South Africa) and the raw data were analysed and aligned to produce a consensus sequence using BioEdit (Ionis Pharmaceuticals, Inc., Carlsbad, CA, USA) followed by BLAST (National Center for Biotechnology Information, Bethesda, MD, USA) analysis. 

The PCR performed on DNA extracted from the cerebrospinal fluid and cyst tissue samples displayed a band of 275 bp on an agarose gel, as expected. The consensus sequence had 97.68% similarity to *T. serialis* 12S ribosomal RNA gene (accession no. KJ490640.1) and was uploaded onto GenBank (accession no. OM501136). 

## 5. Discussion

Coenurosis is a zoonotic disease caused by the larval stages of *Taenia* species, including *T. multiceps*, *T. serialis*, *Taenia brauni*, and *Taenia glomerata* [3]. Canids are the definitive hosts, whilst rodents, horses, and sheep serve as the usual intermediate hosts [3]. Humans become infected by the ingestion of eggs present in the faeces of the definitive hosts. The oncospheres hatch from the eggs into the small intestine, penetrate the intestinal wall, and migrate to target organs, including the brain, eyes, and muscle tissues, through the bloodstream [4]. 

There have been approximately 40 reported cases of human CNS coenurosis, (see Table 1) but, until recently, in all these cases, the assumed etiological agent was confined to *T. multiceps* [3,5]. 

In 2020, however, Yamazawa et al. [6] reported the first case of human CNS coenurosis caused by *T. serialis*. Their case involved a single parenchymal lesion in the occipital lobe, which, following complete surgical excision, was confirmed to be *T. serialis* by mitochondrial gene sequencing.

The case we present is the first case of disseminated subarachnoid coenurosis caused by *T. serialis*. The importance of genetic analysis in accurate diagnosis is underscored by our original misidentification of the organism. Unless a detailed analysis of the arrangement of the parasitic structure is performed, the basic structure of a coenurus at the microscopic level is similar to that of a cysticercus, and histopathological misidentification in not uncommon [3,5,7].

Of interest is the young presentation of this patient, as the majority of the CNS coenurosis cases reported to date [3] have occurred in the adult population. This is not universal, however, with some authors [8] reporting a high prevalence of pediatric age group infections. In all cases, a prolonged exposure to dogs and sheep was documented [3]. A recent study from South Africa [9] found that 50% of their coenurosis cases presented within the pediatric population. 

The spectrum of CNS coenurosis due to *T. multiceps* includes both parenchymal lesions [4,10] and CSF pathway involvement [11]. When buried within the parenchyma, the cyst is surrounded by a thick layer of mononucleated cells and vascular fibrous tissue [10], whereas in the CSF pathways it induces arachnoiditis [12] or diffuse ependymitis [13]. The spectrum of *T. multiceps* CNS coenurosis thus seems to resemble that of the more common infection, NCC, caused by *T. solium* [5]. 

Based on the previous single reported case of intraparenchymal CNS coenurosis and the present case of CSF pathway infection, we speculate that *T. serialis* infection can also produce the two basic pathological forms of NCC [14,15], namely, isolated cysts (cysticercosis cellulosae) and a form macroscopically resembling racemose cysts (cysticercosis racemosus). 

First described by Zenker in 1882, racemose cysticercosis [16] refers to the delicate grape-like bladders of cysts that form within the subarachnoid spaces of the CNS. The exact mechanism by which the cysts proliferate and expand is poorly understood [17]. Although, in an experimental rabbit model, Lachberg et al. [7] were able to induce ‘cysticercosis racemosus’ by *T. serialis* infection, racemose cysticercosis has generally been considered to be exclusively due to *T. solium*, and to our knowledge, to date, only *T. solium* has ever been molecularly proven [18] to result in the racemose form. Now that molecular diagnosis of this form of cestode infection allows for the precise aetiological confirmation [2] of *T. serialis* infection, we postulate that the term coenurosis racemose could be applicable in such cases, if supported by histological evidence of abnormal cyst proliferation.

The radiological differentiation of coenurosis from neurocysticercosis is difficult [19]. Both etiological agents result in cysts that, on MRI, appear dark on T1-weighted images and bright on T2 images [10]. These cysts are suppressed in FLAIR imaging and may have rim-like enhancement [20]. The presence of intra-cystic multiple eccentric nodules visible on T2-weighted imaging, as opposed to isolated eccentric nodules, has been reported to be supportive of coenurosis rather than NCC [19]. On MR spectroscopy, both NCC and coenurosis demonstrate a prominent succinate and a smaller, but still conspicuous, alanine peak [19,21], which helps differentiate them from bacterial abscesses but not from each other. 

Given the limited data on the treatment of CNS *T. serialis* infection, we are of the opinion that the protocols for NCC treatment provide the best evidence for treatment decisions. Some controversy exists as to whether anti-helminthic drugs modify the natural history of NCC [15] and there are concerns regarding the treatment-related exacerbation of disease [22,23]. A recent meta-analysis [24], however, supports the use of cysticidal drugs in parenchymal disease. No controlled trials [22] exist regarding subarachnoid disease; however, consensus opinion [25,26] is that anti-parasitic drugs should be instituted for basal cistern cysticercosis. The optimal duration of anti-parasitic treatment for these lesions is not known, but it is generally felt that therapy should be maintained for longer than in routine parenchymal disease [22,25]. Generally, either praziquantel (50 mg/kg/day) or albendazole (15 mg/kg/day) are used [27,28]. However, the parasiticidal efficacy of these drugs in isolation is poor, with only 30–40% of patients achieving complete parasitic clearance after a first course of treatment [27]. The combination of albendazole and praziquantel [27,29,30], as used in our case, has been shown to be more effective in destroying viable cysts without increased treatment-associated side effects. 

Likewise, the use of corticosteroids therapy, begun prior to antiparasitic drugs, is recommented in NCC treatment [26] to diminish the inflammatory response to treatment, and we followed a similar strategy. 

Along with medical management, once signs of increased intracranial pressure develop, surgical management is normally required. The available treatment options for fourth ventricle NCC include external CSF diversion, typically a VP shunt, suboccipital craniectomy and microsurgical excision, internal CSF diversion (ETV), and endoscopic removal, either alone or in combination [31]. Basal cistern NCC has conventionally been treated with medical management alone but, not infrequently surgical removal is required [32]. Only in rare cases can all the cysts be removed, as they are frequently multiple and adherent to the cranial nerves, brainstem and vasculature due to arachnoiditis [33], with a high morbidity and mortality associated with these procedures [34,35]. Endoscopic options for cisternal NCC have also been employed [32], but these are also limited by the same risks as open techniques [34]. 

As demonstrated in our case, despite the initial adequate control of hydrocephalus, the percentage of shunt complications is high in this group [33,36], with up to 80% requiring shunt revisions in the short term [31,33,36]. 

Likewise, the prognosis of CNS coenurosis is very poor. Until recently, it was essentially a lethal disease [3,10]; however, modern neuroradiological technique resulted in better disease recognition, and patients with isolated parenchymal cysts who undergo surgical resection [3,6,10,37] may survive long-term. Disseminated subarachnoid coenurosis, however, continues to have a very high mortality rate. 

**Table 1 tropicalmed-07-00405-t001:** Reported cases of human coenurosis of the central nervous system.

Author	Age/Gender	CNS Location	Geographical Region	Clinical Outcome	Species
Brumpt, 1913 [38]	40/M	Parenchymal and intraventricular	France	Died	*T. multiceps*
Spencer 1936 (no reference, described by Lescano [3])	Age and gender unknown	Parenchymal	Africa	Unknown	Unknown
Cluver, 1940 [39]	Age and gender unknown	Parenchymal and intraventricular	South Africa	Died	*T. multiceps*
Clapham, 1941 [40]	39/M	Parenchymal and intraventricular	England	Died	*T. multiceps*
Parkinson, 1942 (no reference, described by Lescano [3])	Age and gender unknown	Parenchymal	England	Unknown	Unknown
Roger et al., 1942 [41]	42/F	Parenchymal and cisternal	France	Died	*T. multiceps*
Landells et al., 1949 [42]	14/F	Spinal cord	England	Unknown	*T. multiceps*
Johnstone et al., 1950 [43]	2/M	Parenchymal	USA	Died	*T. multiceps*
Becker et al., 1951 [12]	55/M	Intraventricular	South Africa	Unknown	*T. multiceps*
Becker et al., 1951 [12]	34/M	Cisternal	South Africa	Unknown	*T. multiceps*
Becker et al., 1951 [12]	33/M	Cisternal and intraventricular	South Africa	Unknown	*T. multiceps*
Becker et al., 1951 [44]	28/M	Intraventricular	South Africa	Unknown	*T. multiceps*
Watson et al., 1955 [45]	32/M	Parenchymal	South Africa	Died	*T. multiceps*
Watson et al., 1955 [45]	33/M	Parenchymal	South Africa	Died	*T. multiceps*
Ranque et al., 1955 [46]	Age and gender unknown	Parenchymal	France	Unknown	*T. multiceps*
Bertrand et al., 1956 [47]	Age and gender unknown	Parenchymal	France	Unknown	*T. multiceps*
Correa et al., 1962 [48]	42/F	Cisternal	Brazil	Died	*T. multiceps*
D’Andrea et al., 1964 [49]	Age and gender unknown	Parenchymal	Italy	Unknown	*T. multiceps*
Hermos et al., 1970 [50]	2/M	Parenchymal and spinal cord	USA	Died	*T. multiceps*
Michal et al., 1977 [51]	37/F	Parenchymal and cisternal	Switzerland	Unknown	*T. multiceps*
Jung et al., 1981 [52]	48/M	Parenchymal	USA	Survived	Unknown
Schellhas et al., 1985 [11]	3/F	Parenchymal, intraventricular and spinal cord	USA	Died	*T. multiceps*
Pau et al., 1987 [10]	54/M	Parenchymal and cisternal	Italy	Survived	*T. multiceps*
Pau et al., 1990 [53]	28/M	Parenchymal	Italy	Survived	*T. multiceps*
Pau et al., 1990 [53]	51/F	Cisternal	Italy	Survived	*T. multiceps*
Pau et al., 1990 [53]	32/M	Parenchymal	Italy	Survived	*T. multiceps*
Malomo et al., 1990 [54]	Age and gender unknown	Parenchymal	Nigeria	Unknown	*T. multiceps*
Sabattani et al., 2004 [55]	46/F	Parenchymal	Italy	Unknown	*T. multiceps*
El-On et al., 2008 [4]	4/F	Parenchymal	Israel	Survived	*T. multiceps*
Mahadevan et al., 2011 [56]	55/M	Parenchymal	India	Survived	*T. multiceps*
Mahadevan et al., 2011 [56]	36/M	Parenchymal	India	Survived	*T. multiceps*
Ali et al., 2019 [37]	63/M	Base of skull with parenchymal extension	India	Survived	*T. multiceps*
Kulanthaiveluet al., 2020 [19]	28/M	Parenchymal	India	Survived	*T. multiceps*
Kulanthaiveluet al., 2020 [19]	55/M	Parenchymal	India	Survived	*T. multiceps*
Kulanthaiveluet al., 2020 [19]	24/F	Intraventricular	India	Survived	*T. multiceps*
Kulanthaiveluet al., 2020 [19]	56/F	Parenchymal	India	Survived	*T. multiceps*
Kulanthaiveluet al., 2020 [19]	50/M	Parenchymal	India	Survived	*T. multiceps*
Kulanthaiveluet al., 2020 [19]	34/M	Parenchymal	India	Survived	*T. multiceps*
Yamazawa et al., 2020 [6]	38/M	Parenchymal	Japan	Survived	*T. serialis*
Nhlonzi et al., 2022 [9]	46/F	Parenchymal	South Africa	Died	*T. multiceps*
Present case	5/M	Cisternal and intraventricular	South Africa	Died	*T. serialis*

## 6. Conclusions

Traditionally, human coenurosis has been ascribed to *T. multiceps* [10], but the increasing availability of molecular methods is likely to improve the accuracy of the identification of larval cestode species. This report highlights only the second case of CNS coenurosis, and the first description of disseminated subarachnoid coenurosis, caused by *T. serialis*. Unfortunately, regardless of the pathogen identity, this form of CNS larval cestode infection is difficult to treat satisfactorily. 

## Figures and Tables

**Figure 1 tropicalmed-07-00405-f001:**
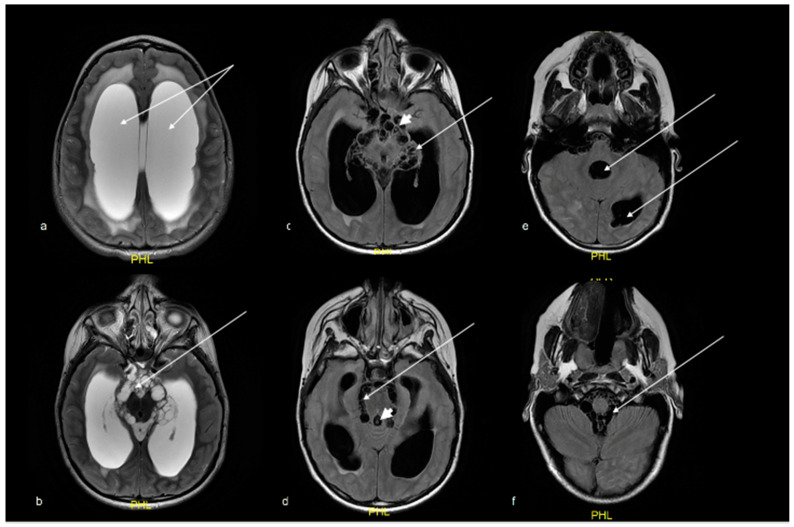
(**a**) T2 axial MRI brain demonstrates dilated lateral ventricles (arrows) with transependymal oedema in keeping with acute hydrocephalus. (**b**) T2 axial MRI and (**c**–**f**) FLAIR axial MRI brain demonstrates multiple cysts (arrows) in the basal cisterns, cerebellomedullary cistern, pontomedullary cistern, fourth ventricle and left posterior horn of lateral ventricle. Some cysts demonstrated a central dot in keeping with a scolex of NCC (arrowhead).

**Figure 2 tropicalmed-07-00405-f002:**
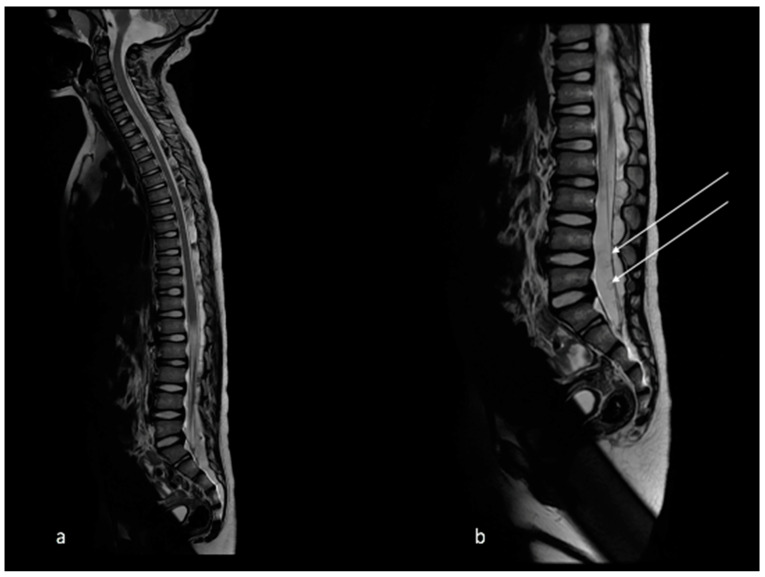
T2 sagittal MRI spine demonstrates (**a**) multiple cysts in the posterior spinal epidural space extending from lower cervical to the lumbar region with (**b**) smaller cysts within the subarachnoid space around filum terminale (arrows).

**Figure 3 tropicalmed-07-00405-f003:**
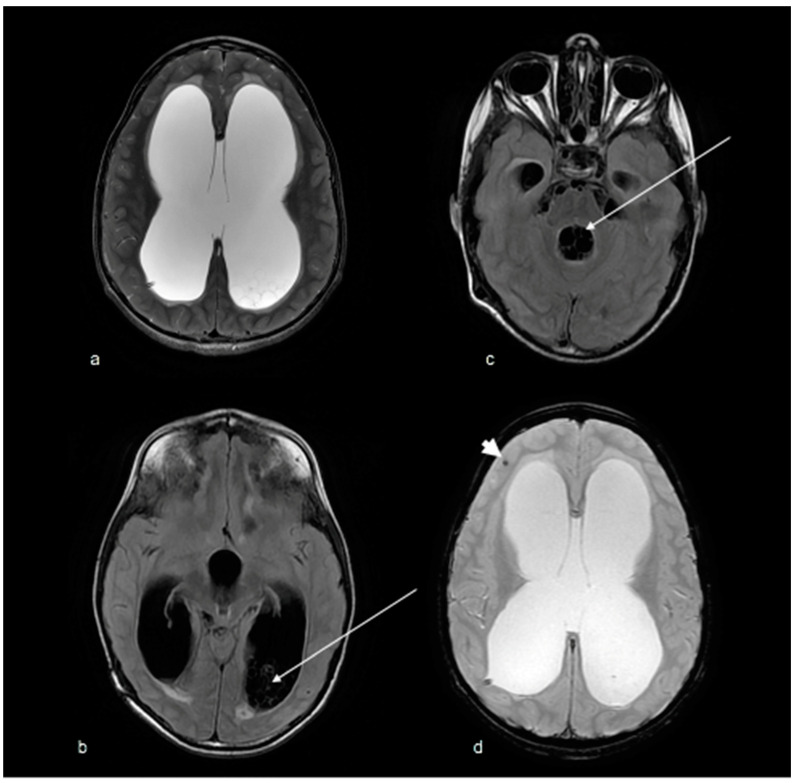
(**a**) T2 axial MRI and (**b**–**c**) FLAIR axial MRI brain demonstrates acute hydrocephalus with a re-distribution of cysts (arrows) within the CSF spaces. (**d**) GRE axial MRI brain demonstrates a new isolated right frontal lobe lesion (arrowhead) that blooms on GRE, most likely in keeping with a calcified granuloma.

**Figure 4 tropicalmed-07-00405-f004:**
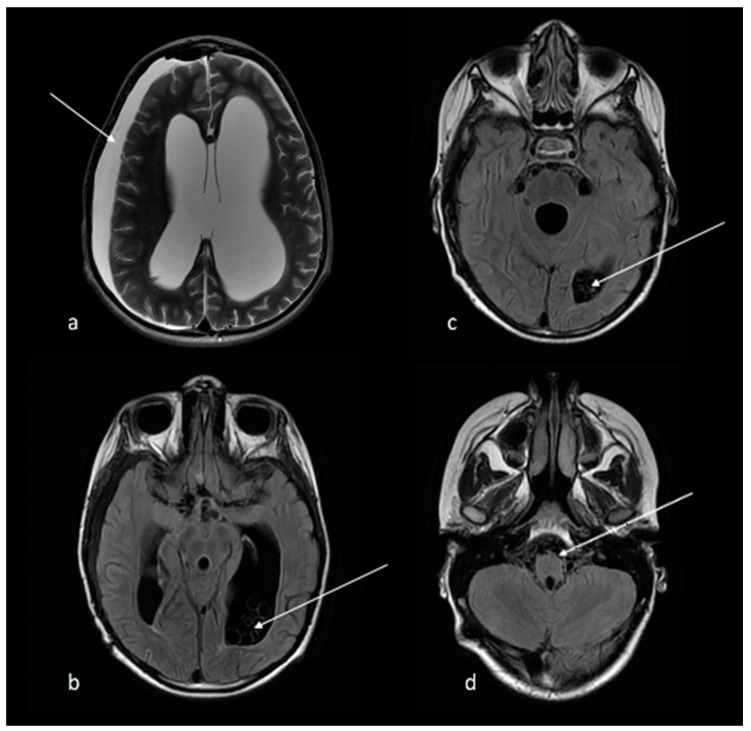
(**a**) T2 axial MRI brain demonstrates a right subdural collection (arrow) and further improvement in the hydrocephalus. (**b**–**d**) FLAIR axial MRI brain shows improvement in number and size of cysts (arrows), especially in the basal cisterns, with cysts no longer visualized in in the 4th ventricle.

## Data Availability

Pathogen sequence data is available on GenBank, accession no. OM501136 (https://www.ncbi.nlm.nih.gov/genbank/).

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
