# Peer review of "Disseminated Human Subarachnoid Coenurosis"

_tropicalmed, 2022, doi:10.3390/tropicalmed7120405_

Round 1

Reviewer 1 Report

Dear authors, I have received an interesting case report regarding "Disseminated Human Subarachnoid Coenurosis." I suggest amending a few minor points before publication:

Abstract:

The last sentence does not feel right in the abstract, especially ending the abstract with a question. Please rephrase it.

Figures:

A general remark about the figures is that the statistics should be appropriately labeled to pinpoint where the pathologies are detected. The authors also did not mention figure (a) anywhere in the labeling of figures.

Case Report:

- Why wasn't the patient followed up after the first surgery and waiting for the patient to deteriorate after three months?

- After how many LOS was the patient discharged?

- Why weren't genetic studies conducted in the first surgery?

- Why wasn't the biopsy done at all?

Discussion:

- The authors should discuss more MRI findings of coenurosis in the brain and spine

- What is the mortality rate of patients diagnosed with the same diagnosis?

- Please provide a more straightforward explanation of why the child suffers from this disease endemicity in this area. Bear in mind that he is only five years old; thus, the transmission mode could be explained in great detail.

Author Response

Dear reviewer Thank you for the review. Please see attached document for a reply to your comments.

Reviewer 2 Report

It is an interesting case report since human cenurosis is a neglected disease. Moreover, it shows how the molecular techniques should be the gold standard for all the laboratories in order to improve diagnosis and unveil new parasitic diseases.

I recommend authors to do a review of all the cases of this disease in order draw some conclusions and be useful to other clinicians

Author Response

Dear Reviewer 2 

Thank you very much for your review.

Please find attached a reply to your suggestions, including the summary you required.

Regards

DR JJ Labuschagne 

Reviewer 3 Report

The manuscript is original and I believe that contained valuable data for authors. I show some minor comments on the text. After corrections the manuscript can be accepted for publication.

Author Response

Dear Reviewer 3

Thank you for the favourable review of our manuscript.

We have made the minor changes as suggested by you.

Regards

Dr JJ Labuschagne

Reviewer 4 Report

I did read this report with enthusiasm but unfortunately I see two major issues on this submission.  FIrst, parasite material was extracted on surgery.  It follows that a histological demonstration of the nature of the coenurus must have been seen on histology or even at macroscopy - and these are absent in the report. On the report itself, I feel that a single marker in the absence of supportive histology does not provide enough certainty of the species diagnosis

Author Response

Dear reviewer 4

Thank you very much for the review of our manuscript.

Please find attached a reposes to your suggestions.

Regards

DR JJ Labuschagne 
